# Each Complexity Deserves a Pruning Policy

**Hanshi Wang**[1,2,3,5]*, **Yuhao Xu**[3], **Zekun Xu**[1,2], **Jin Gao**[1,2,5]†,
**Yufan Liu**[1,2,5], **Weiming Hu**[1,2,5,6], **Ke Wang**[7], **Zhipeng Zhang**[3,4]†

[1]State Key Laboratory of Multimodal Artificial Intelligence Systems (MAIS), CASIA
[2]School of Artificial Intelligence, University of Chinese Academy of Sciences
[3]AutoLab, School of Artificial Intelligence, Shanghai Jiao Tong University [4]Anyverse Intelligence
[5]Beijing Key Laboratory of Super Intelligent Security of Multi-Modal Information
[6]School of Information Science and Technology, ShanghaiTech University [7]KargoBot
{hanshi.wang.cv, zhipeng.zhang.cv}@outlook.com; jin.gao@nlpr.ia.ac.cn

## Abstract

The established redundancy in visual tokens within large vision–language models (LVLMs) allows for pruning to effectively reduce their substantial computational demands. Empirical evidence from previous works indicates that visual tokens in later decoder stages receive less attention than shallow layers. Then, previous methods typically employ heuristics layer-specific pruning strategies where, although the number of tokens removed may differ across decoder layers, the overall pruning schedule is fixed and applied uniformly to all input samples and tasks, failing to align token elimination with the model's holistic reasoning trajectory. Cognitive science indicates that human visual processing often begins with broad exploration to accumulate evidence before narrowing focus as the target becomes distinct. Our experiments reveal an analogous pattern in LVLMs. This observation strongly suggests that neither a fixed pruning schedule nor a heuristics layer-wise strategy can optimally accommodate the diverse complexities inherent in different inputs. To overcome this limitation, we introduce Complexity-Adaptive Pruning (AutoPrune), which is a training-free, plug-and-play framework that tailors pruning policies to varying sample and task complexities. Specifically, AutoPrune quantifies the mutual information between visual and textual tokens, and then projects this signal to a budget-constrained logistic retention curve. Each such logistic curve, defined by its unique shape, is shown to effectively correspond with the specific complexity of different tasks, and can easily guarantee adherence to a pre-defined computational constraints. We evaluate AutoPrune not only on standard vision-language tasks but also on Vision-Language-Action (VLA) models for autonomous driving. Notably, when applied to LLaVA-1.5-7B, our method prunes 89% of visual tokens and reduces inference FLOPs by 76.8%, but still retaining 96.7% of the original accuracy averaged over all tasks. This corresponds to a 9.1% improvement over the recent work PDrop (CVPR'2025), demonstrating the effectivenes. Code is available at `https://github.com/AutoLab-SAI-SJTU/AutoPrune`.

## 1  Introduction

Vision and Language Models (VLMs) have rapidly emerged as the backbone of modern multimodal systems, powering tasks such as image captioning [1, 2, 3], visual question answering (VQA) [4, 5] and multimodal dialogue [6, 7]. Recent extensions to embodied intelligence like the wildly deployed autonomous driving system, exemplified by Vision–Language Action frameworks [8, 9, 10], further couple these perceptual capabilities with driving control, permitting end-to-end reasoning. However,

---

*This work was completed during Hanshi's remote internship at SJTU and co-mentored by Prof. Zhipeng Zhang. †Corresponding author.

39th Conference on Neural Information Processing Systems (NeurIPS 2025).

tokenizing high-resolution images or video for LLMs yields excessively long visual sequences, which in turn create memory and latency bottlenecks and make efficient pruning essential for real-time use. Among the diverse methods aimed at boosting the efficiency of VLMs, training-free token pruning stands out as a significant technique because of its simplicity [11, 12, 13, 14, 15, 16, 17].

A review of related literature reveals a prevailing understanding that the informational contribution of visual tokens substantially diminishes during the later stages of the VLM decoder [18]. Existing training-free token pruning methods reflecting this principle typically adhere to predetermined fixed pruning schedules [19, 13] or alternatively they employ layer specific heuristics [20, 21] yet without explicit adherence to a global computational budget. However for reasoning intensive tasks which necessitate iterative inference and dynamic cross modal fusion, such fixed pruning policies lack adaptability and cannot meet the sample specific and task specific demands. Our experiments in Sec. 3.2 underscore this limitation showing that saliency patterns and consequently token importance vary significantly with the input image and the posed query. While certain layer wise heuristics attempt to prune tokens differently based on factors like decoder layer depth, their handcrafted nature presents challenges as these designs often fail to guarantee adherence to a target token count or FLOPs budget without extensive manual tuning nor do they provide robust evidence of generalizability across diverse scenarios. This context therefore naturally motivates a critical question that *"Is it feasible to develop a pruning methodology that ♠ dynamically adjusts to the varying complexities of individual samples and tasks while ♡ readily adhering to a predefined computational budget and ♣ concurrently upholding principles of simplicity and broad generalizability?"*

Since humans excel at complex visuolinguistic reasoning, we tend to find the answer from neuroscience first. The studies in which reveals that for clearly expressions (*simple samples and tasks*), ventral visual stream and temporal-language areas rapidly converge on the referent, yielding single sustained fixations [22, 23, 24]. In contrast, ambiguous or indirect descriptions (*complex samples and tasks*) engage dorsolateral prefrontal and parietal networks, with the prefrontal cortex maintaining competing hypotheses while the dorsal attention system drives iterative gaze shifts [25, 26]. Mirroring this exploration–exploitation cycle, our analysis of VLMs in Fig. 1 also proves that *simple samples and tasks* induce a rapid collapse of cross-modal attention within early layers, whereas *complex samples and tasks* sustain diffuse attention and exhibit pronounced inter-layer saliency fluctuations. These observations demonstrate that fixed pruning schedules, whether aggressive or conservative, cannot satisfy the varied demands of reasoning.

Leveraging these insights, our work primarily proposes Complexity-Adaptive Pruning (AutoPrune), a framework that endows each input with an individualized pruning policy. To achieve this goal, we believe the core challenge lies in quantifying sample complexity in a manner and representing the latent thought process compatible with training-free deployment and adherence to a fixed computational budget. Drawing on neuroscientific evidence that tightly coupled cross-modal signals shorten human reasoning paths, we measure the *mutual information* between early-layer visual and textual tokens to identify input complexity. A high value implies a simple sample with an easily localized answer, while a low value flags a complex sample requiring broader exploration. We map this scalar complexity estimate onto *logistic retention curves* that mimic the human explore-commit-stabilise pattern observed in eye-tracking studies. Each curve represents a distinct pruning policy, where the values at different points along the curve dictate the degree of token pruning at varying depths of the decoder. The curve's slope and inflection point are modulated linearly by the mutual information score, yielding aggressive front-loaded pruning for simple samples and conservative, late-onset pruning for complex ones. To guarantee the pre-defined cost, we analytically integrate each curve, rescaling it so that the area under the curve equals a user-specified token or FLOPs budget.

Due to its simple and plug-and-play architecture, our AutoPrune can be seamlessly integrated into a variety of VLM and VLA models, including LLaVA-1.5 [27], LLaVA-NeXT [28], and Senna [29] for autonomous driving. Experiments on standard vision–language benchmarks, autonomous driving scene understanding and planning demonstrate that AutoPrune consistently outperforms existing training-free methods across a broad range of pruning ratios. For instance, when applied to LLaVA-1.5-7B, AutoPrune prunes 89% of visual tokens and reduces inference FLOPs by 76.8%, but still retaining 96.7% of the original accuracy averaged over all tasks. This corresponds to a 9.1% improvement over the recent work PDrop (CVPR2025) [13].

Our contributions include: ♠ We present a cognitive neuroscience–inspired analysis that systematically links sample and task complexity with token retention decay and inter-layer fluctuations

in cross-modal attention. ♡ We propose AutoPrune, a training-free complexity-adaptive pruning framework that computes mutual information from visual–textual attention and maps it to a budget-constrained logistic retention schedule, assigning each sample and task a customized pruning curve under any specified token or FLOPs budget. ♣ We demonstrate the generality of our approach by integrating AutoPrune into multiple VLM and VLA models and benchmarking against diverse baselines. Extensive experiments show that our method consistently outperforms prior state-of-the-art approaches across various tasks and reduction ratios.

## 2 Related Work

**Vision-Language Models (VLMs).** VLMs have achieved significant progress in integrating visual and textual modalities, enabling sophisticated tasks such as image captioning [1, 2, 3], visual question answering (VQA) [4, 5], and multimodal dialogue [6, 7]. Their broad world knowledge has spurred embodied applications and led to VLA models [8, 9], which add action generation for control, with autonomous driving as a representative application. A typical design uses a visual encoder for features and an LLM for multimodal reasoning and output. This pairing grants visual perception but expands inputs into long token sequences. High resolution images [30] and video [31, 32] amplify memory and latency. Consequently, optimizing the inference efficiency of these powerful models is a critical prerequisite for their practical deployment in resource-constrained real-world scenarios. Among the diverse methods aimed at boosting the efficiency of VLMs, token pruning stands out as a significant technique, broadly divisible into training-based and training-free paradigms.

**Token Pruning.** In pursuit of task-optimized efficiency, one prominent line of research focuses on training-based pruning methodologies. These approaches necessitate supplementary training or fine-tuning stages to instill task-specific pruning behaviors, potentially enhancing performance metrics on target applications [11, 33, 34, 12, 35]. Training-free pruning avoids retraining and can be applied directly to pretrained models [18]. Methods are commonly grouped by pruning stage. Pre decoder pruning selects a subset of visual tokens with unsupervised similarity or lightweight scores before the LLM, as in TopV [36] and FasterVLM [19]. Intra decoder pruning removes tokens during inference across LLM layers using preset layerwise schedules or attention statistics, as in PyramidDrop [13] and ZipVL [14]. Despite their appeal and diverse application points (pre- or intra-decoder), a critical unresolved shortcoming persists in existing training-free methods. Specifically, regardless of the pruning stage, the vast majority employ fixed pruning policies [11, 12, 13, 14, 15, 16, 17]. The rigidity of such static approaches proves problematic. Our experimental observations within VLMs reveal that, even for the same input image, the pattern of token importance varies dynamically across decoder layers depending on the specific question posed. Consequently, a fixed pruning policy is inherently ill-suited to adapt to these variations that are contingent on both input query and processing depth within decoder, underscoring the need for more adaptive, context-aware pruning strategies.

## 3 Method

### 3.1 Preliminaries

We cast token pruning as a constrained optimisation problem whose decision variables specify 1) **how many** visual tokens survive in each transformer block, 2) **how to select** specific tokens for retention, and 3) **how to revive** discarded tokens. All three decisions are optimised jointly under a global constraint on total computation. Concretely, we denote the token-allocation policy as $\xi$ specifying the number of tokens preserved at each layer $i$, the token-selection policy as $\pi$ governing token retention, and the token-revival policy as $\rho$ dictating how discarded tokens are revived and remapped. For a data distribution $\mathcal{D}$ and task loss $\ell$, the expected loss is defined as:

$$\mathcal{L}(\xi, \pi, \rho) = \mathbb{E}_{(\mathbf{V}, \mathbf{T}, \mathbf{y}) \sim \mathcal{D}} \, \ell\big(\mathbf{y}, \, f_\theta(\mathbf{V}, \mathbf{T}; \xi, \pi, \rho)\big), \tag{1}$$

where $\mathbf{V} \in \mathbb{R}^{N_v \times d}$ and $\mathbf{T} \in \mathbb{R}^{N_t \times d}$ are the image and text tokens, $\mathbf{y}$ is the ground-truth, and $f_\theta$ is the vision–language model. We aim to minimise $\mathcal{L}$ subject to a global compute budget $c_{max}$:

$$\min_{\xi, \pi, \rho} \mathcal{L}(\xi, \pi, \rho) \quad \text{s.t.} \quad \sum_{i=1}^{L} c_i\big(\xi, \rho\big) \leq c_{max}, \tag{2}$$

where $c_i(\xi, \rho)$ measures the computational cost incurred by pruning and potential revival at layer $i$.

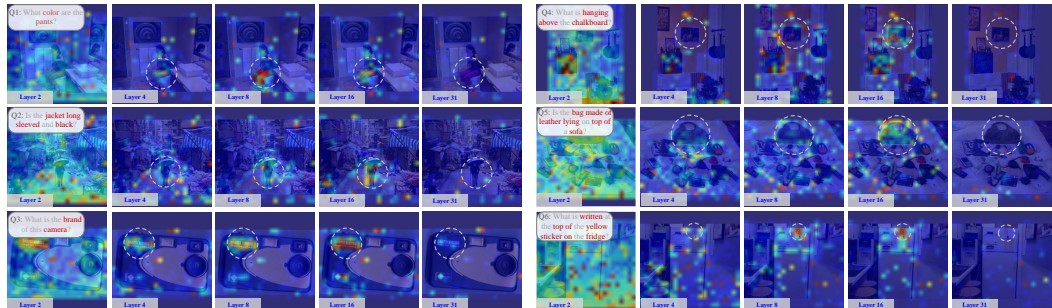

Figure 1: **Layer-wise Visual–Textual Interaction Patterns.** By visualizing cross-modal attention at layers 2, 4, 8 and 16 of the VLM, we observe that for tasks requiring only object identification, attention rapidly converges on the salient region and remains stable, whereas for reasoning-intensive tasks attention shifts progressively across layers.

We focus on optimizing the token-allocation policy $\xi$, which governs how many tokens are preserved at each layer. Prior approaches fall into two camps: 1) a uniform pruning schedule applied identically across all tasks, which cannot adapt to varying visual–textual demands, and 2) per-layer schemes that tune pruning independently but lack a mechanism to enforce a global compute budget, often resulting in insufficient pruning and limited speedup. In contrast, our method dynamically allocates token budgets in a global manner which rigorously satisfying the overall computation constraint, thereby unifying adaptability and acceleration.

## 3.2 Neuroscience Inspiration and Analysis

Neuroscientific research [22, 23, 24, 25, 26] shows that the neural resources engaged in visuolinguistic processing scale with task complexity. When the text unambiguously specifies a target (*simple tasks*), object-selective regions in the ventral visual stream cooperate with temporal-language areas to form a rapid, stable representation, and attention remains anchored to the relevant image region. Indirect or ambiguous links (*complex tasks*) elicit additional activity in dorsolateral prefrontal and parietal control networks, the prefrontal cortex maintains competing interpretations while the dorsal attention system reallocates gaze among candidate referents. Eye-tracking corroborates this shift, revealing single, sustained fixations in simple conditions but iterative scans in complex ones, which provides evidence of top-down guidance by language riven inference.

Guided by these neuroscience cognitive insights, we conduct a fine-grained analysis of cross-modal attention in vision-language models. As illustrated in Fig. 1, we analyze the model's behavior on both simple and complex tasks and identify two key findings.

**Task-Sample Adaptive Token Number Decay.** Fig. 1 (left) demonstrates that in simple tasks and samples (here in this example, "task" indicates the asked question, "sample" denotes the input image), where the referring expression unambiguously identifies the target, cross-modal attention outside the relevant region collapses within the first few layers. At that stage, only tokens corresponding to the target remain active. In contrast, as shown in Fig. 1(right) , for complex tasks and samples that demand non-trivial visual inference, attention over image tokens decays gradually and remains widely dispersed in the initial layers, indicating the model's uncertainty about where pertinent evidence resides. Hence, simple tasks and samples permits aggressive pruning at shallow depths, whereas reasoning-intensive prompts benefit from postponing token removal until deeper layers. These findings motivate a task-adaptive pruning policy that dynamically models global token trajectories, preserving a wide set of tokens when alignment is ambiguous and confidently discarding irrelevant tokens at shallow depths when alignment is clear. However, as discussed before, existing methods cannot simultaneously capture both the gloabl token trajectories and the compute budget, limiting their ability to reconcile adaptivity with efficiency.

**Inter-Layer Saliency Position Variation.** In *simple tasks*, cross-modal attention converges by an early layer and remains stable thereafter (Fig. 1, left), indicating that further exploratory inference provides no additional benefit. In *complex tasks*, the saliency of individual visual tokens varies across layers. Specifically, as shown in Fig. 1(right line 1), the model initially attend to a chalkboard, shift focus to surrounding regions in intermediate layers, and return to the true target by layer 16. These fluctuations reveal an intrinsic search mechanism that probes alternative regions under weak initial

cues and progressively refines attention as higher-order features emerge. In summary, for simple tasks, aggressive pruning can be applied once attention has converged, for complex tasks, maintaining a larger token set across layers is essential to support ongoing inference and resolve ambiguity.

These findings indicate that effective pruning must follow a dynamically modeled, globally coherent trajectory. However, existing approaches either lack a mechanism to capture such dynamics or fail to achieve global pruning trajectory. Our framework employs complexity-aware pruning that adapts to each sample, task, and layer saliency position variation to produce a dynamic, globally consistent trajectory within a fixed computational budget.

### 3.3 Complexity-Adaptive Pruning

Then the next questions are: *(1) how to formulate a reliable indicator for assessing the sample and task complexity?* and *(2) how to translate this indicator into a policy that is not only formally simple but also allows for straightforward management of the overall computational budget?* In pursuing an answer, we again drew inspiration from neuroscience.

Neuroscientific evidence indicates [37] that in semantically congruent audiovisual contexts information flow from early visual areas to language integration regions is both stronger and more direct, reflecting enhanced bidirectional coupling. In incongruent contexts higher order regions such as the prefrontal cortex are recruited to resolve the mismatch, which attenuates direct exchange between lower level sensory areas. This contrast implies that the extent of information exchange in the initial layers can serve as a proxy for complexity. To quantify this, **for the first question above**, we compute the mutual information between visual tokens and textual tokens. Specifically, high mutual information denotes a direct, "simple" task that allows aggressive pruning, whereas low mutual information signifies an indirect, "complex" task that demands conservative token retention.

Extensive eye-tracking and electroencephalography researches [25, 38, 39, 40] show that, in the image-based question-answering tasks, the time course of human fixations is well described by a logistic (*S-shaped*) curve. Specifically, an initial epoch of broadly distributed gaze is followed by a steep rise in target-fixation probability once task-relevant evidence exceeds a cognitive threshold, after which fixations stabilise on a small region of the scene. This explore–commit–stabilise pattern appears in purely visual settings and in cross-modal variants that combine visual scenes with spoken or multisensory cues, indicating a modality-general principle of attention allocation. Inspired by this finding, **for the second question above**, we define a logistic retention function that emulates a human-like, iterative inspection process, applying aggressive early pruning in simple tasks with strong alignment to isolate key tokens and reserve budget for deeper analysis, whereas in complex tasks we prune conservatively at first to avoid discarding critical information prematurely. Besides the neurological explanation, we select the logistic function due to its inherent simplicity and the facility with which its shape can be modulated by adjusting hyperparameters. These hyperparameters can be efficiently derived from the indicator we introduce. As illustrated in Fig. 2, steeper slopes indicate lower mutual information and correspond to lower sample scores, thereby validating our method.

Subsequently, we elaborate on these two essential components.

**Mutual Information for Cross-Modal Alignment.** To translate the qualitative insights from our neuroscientific analysis into a quantitative signal that can steer pruning, we require a scalar measure of how tightly a textual prompt constrains the visual scene. Mutual information naturally fulfils this role because it captures the reduction in visual uncertainty provided by the text and is directly computable from cross-modal attention. Specifically, we quantify the mutual information [41] between the visual tokens $\mathbf{V}$ and textual tokens $\mathbf{T}$ by

$$I(\mathbf{V}, \mathbf{T}) = \sum_{i=1}^{N_\mathrm{v}} \sum_{j=1}^{N_\mathrm{t}} p\big(v_\mathrm{i}, t_\mathrm{j}\big) \, \log \frac{p\big(v_\mathrm{i}, t_\mathrm{j}\big)}{p(v_\mathrm{i}) \, p(t_\mathrm{j})}, \tag{3}$$

where $N_\mathrm{v}$ and $N_\mathrm{t}$ denote the numbers of visual and textual tokens, respectively. We estimate the joint and marginal probabilities by interpreting the transformer's softmax-normalized attention weights $\alpha_\mathrm{ji}$ (from text token $t_\mathrm{j}$ to visual token $v_\mathrm{i}$) as probabilities $p(v_\mathrm{i} \mid t_\mathrm{j})$ under a uniform text prior, yielding

$$p\big(v_\mathrm{i}, t_\mathrm{j}\big) = \frac{1}{N_\mathrm{t}} \, \alpha_\mathrm{j,i}, \quad p(v_\mathrm{i}) = \sum_{j=1}^{N_\mathrm{t}} p\big(v_\mathrm{i}, t_\mathrm{j}\big), \quad p(t_\mathrm{j}) = \frac{1}{N_\mathrm{t}}. \tag{4}$$

This approach leverages the fact that attention's softmax outputs form a valid distribution [42, 43], allowing direct computation of mutual information from the attention maps.

As shown in Fig. 2, a large value of $I(\mathbf{V}, \mathbf{T})$ indicates that the textual prompt sharply constrains the space of plausible visual interpretations, allowing the model to confidently localize the relevant region early and perform aggressive front-loaded pruning of non-essential visual tokens (orange curve). Conversely, a small mutual information means weak or indirect correspondence, so the network must preserve multiple visual hypotheses across layers, leading to a gradual, unstable reduction of attention (blue curve) [44, 45]. Thus, mutual information provides a principled scalar proxy for task-specific text–vision alignment, guiding the pruning schedule introduced in this work.

**Budget-Constrained Logistic Retention.** For a question–answer pair $q$, the policy is defined as,

$$f_{\mathrm{q}}(x) \;=\; \frac{N_{\mathrm{init}}}{1 + \exp\big(k_{\mathrm{q}}(x - x_0^{\mathrm{q}})\big)} \tag{5}$$

where $N_{\mathrm{init}}$ is the initial token count, $k_{\mathrm{q}}$ controls the steepness of the retention decay, and $x_0^{\mathrm{q}}$ denotes the layer at which the retention rate falls to half of $N_{\mathrm{init}}$.

Practical deployment requires an fixed compute budget $C_{\mathrm{max}}$, prior methods approximate this cost by reporting the average per-layer token number [20, 13]. Based on their method and to meet requirement of compute budget we integrate $f(x)$ over the depth domain $[0, L]$,

$$F_{\mathrm{q}}(x) \;=\; \int f_{\mathrm{q}}(x)\,dx \;=\; N_{\mathrm{init}}\left[(x - x_0^{\mathrm{q}}) - \frac{1}{k_{\mathrm{q}}}\ln\big(1 + \mathrm{e}^{k_{\mathrm{q}}(x - x_0^{\mathrm{q}})}\big)\right] \tag{6}$$

and compute the area $I_{\mathrm{q}} = F_{\mathrm{q}}(L) - F_{\mathrm{q}}(0)$. We then renormalise the curve by

$$\hat{f}_{\mathrm{q}}(x) \;=\; \frac{c_{\mathrm{max}}/I_{\mathrm{q}}}{1 + \exp\big(k_{\mathrm{q}}\,(x - x_0^{\mathrm{q}})\big)} \tag{7}$$

so that $\int_0^L \hat{f}_{\mathrm{q}}(x)\,dx = c_{\mathrm{max}}$. This procedure preserves the shape of curve while guaranteeing identical computational complexity across tasks. Since network layers are discrete, and token counts must be integer-valued. We therefore evaluate $\hat{f}_{\mathrm{q}}(i)$ at each layer $i \in \{0, \dots, L\}$ to the nearest integer, and adjust a global scale factor $s$ until $\sum_i^L \lfloor s \cdot \hat{f}_{\mathrm{q}}(i) \rfloor \simeq c_{\mathrm{max}}$ with binary search. The same procedure applies when the budget is expressed in FLOPs rather than token counts. Let $c(x)$ denote the per-layer cost incurred by retaining $\hat{f}_{\mathrm{q}}(x)$ tokens. Replacing token number by $\int_0^L c\big(\hat{f}_{\mathrm{q}}(x)\big)\,dx$ and rescaling as above yields a schedule that respects any desired FLOPs target.

**Dynamic Logistic Pruning Policy.** We treat the mutual information between visual and textual tokens as a measure of alignment strength that distinguishes simple tasks from complex ones. When mutual information is high, indicating strong correspondence between modalities,

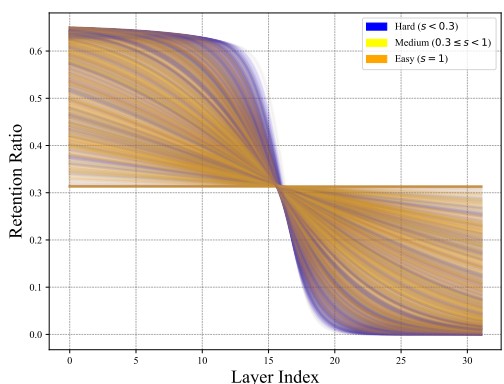

Figure 2: Logistic retention curves on the TextVQA dataset. Each curve corresponds to a QA pair, and is parameterized by the mutual information between visual and textual tokens. Samples/Tasks exhibiting lower mutual information show more conservative retention.

we configure the logistic retention function to decline rapidly in the early layers, enabling the model to prune surplus tokens in easy tasks (orange curve). Conversely, when mutual information is low, signifying weaker alignment, we maintain a prolonged plateau in the logistic curve and defer its sharp descent until later layers, thereby preserving a higher token budget to safeguard critical evidence in demanding tasks (blue curve).

We implement this effect by letting both the slope $k$ and the inflection point $x_0$ depend linearly on the mutual information. Concretely, we set

$$k_{\mathrm{q}} = k_0 - \gamma\, I_{\mathrm{q}}(\mathbf{V}, \mathbf{T}), \quad x_0^{\mathrm{q}} = x_0 + \beta\, I_{\mathrm{q}}(\mathbf{V}, \mathbf{T}), \tag{8}$$

where $k_0, \gamma, \beta > 0$ and $x_0$ is the given params. Hence, a small $I_q(\mathbf{V}, \mathbf{T})$ produces a larger slope $k_q$ and a higher $x_0^q$, thereby retaining more tokens prior to the inflection point to avoid prematurely discarding critical information. In contrast, a large $I_q(\mathbf{V}, \mathbf{T})$ reduces the number of tokens preserved before $x_0^q$, enabling the model to repeatedly concentrate its computations on the most salient features.

### 3.4 Theoretical Analysis of Computational Complexity

To evaluate the efficiency of our pruning algorithm, we derive its overall time complexity as

$$\mathcal{O}\big(N_h\,N_t\,N_v \;+\; N_v\,\log(N_v)\,L \;+\; L\,N_v\,\log(N_v)\big) \;\approx\; \mathcal{O}\big(N_h\,N_t\,N_v\big), \tag{9}$$

where $N_h$, $N_t$ and $N_v$ denote the numbers of attention heads, textual tokens and visual tokens, respectively, and $L$ is the number of layers. The first term $N_h\,N_t\,N_v$ corresponds to mutual-information estimation, the second term $N_v\,\log(N_v)\,L$ reflects the generation of the logistic function and normalization, and the third term $L\,N_v\,\log(N_v)$ captures per-layer token sorting. None of these operations depends on feature dimension $d$ (*e.g.* $d = 4096$), and under typical settings ($N_h = 32$, $N_t, N_v \approx$ several hundred, $L = 32$) the additional overhead is negligible compared with overall inference cost.

## 4 Experiments

We evaluate our framework across a diverse suite of vision–language benchmarks, comparing it against state-of-the-art token pruning methods on a single NVIDIA Tesla A100 GPU. Tab. 1 reports results on multi-modal tasks commonly employed in previous token pruning studies. Tab. 2 assesses the generalizability of our work by applying our AutoPrune to other VLMs. Furthermore, we demonstrate the generalizability of AutoPrune to embodied robots for autonomous driving (Tab. 3). Notably, the core AutoPrune pipeline and even hyper-parameters remains unaltered when applied to the embodied task, thereby clearly and fairly demonstrating the broad applicability of our work.

### 4.1 Results with LLaVA

We evaluate our AutoPrune integrated into LLaVA-1.5-7B [27] on five standard vision-language benchmarks, including MME [46], MMB [47], ScienceQA (SQA) [48], GQA [49], and TextVQA [50]. As shown in Tab. 1, our AutoPrune consistently outperforms all competitors across the entire sparsity spectrum and exhibits graceful degradation as the visual token budget diminishes. At an aggressive pruning rate of 89% (retaining only 64 tokens), AutoPrune maintains 96.7% of the full-model accuracy, whereas the strongest baseline (PDrop [13] in CVPR'2025) achieves only 87.6%. Under moderate pruning (78% removal, 128 tokens), AutoPrune preserves 98.1% of original performance, compared to 95.6% for PDrop and under 93% for other methods. At a pruning level of 66% removal (192 tokens), AutoPrune achieves virtually lossless performance by maintaining 99.0% accuracy while reducing FLOPs by over 57%. These results demonstrate that our complexity-adaptive pruning schedule not only attains the highest absolute accuracy at all pruning levels but also minimizes performance degradation as the token budget decreases.

### 4.2 Results with LLAVA-NeXT

To validate the generality of our approach, we evaluate its performance on LLaVA-NeXT-7B [28] as detailed in Tab. 2, utilizing three distinct token budgets (640, 320, and 160). For equitable comparison, all methods are benchmarked on datasets employed in prior work [19], encompassing VQA$^{V2}$ [52], GQA [49], TextVQA [50], POPE [53], and MME [46]. When retaining 320 tokens, our method maintains a relative performance retention of 98.2%, outperforming all compared methods. Under the most stringent budget of 160 retained tokens, our approach preserves 94.9% of its original performance, exceeding the nearest competitor FasterVLM (86.7%) by more than five percentage points. These results affirm the robustness of our pruning strategy across diverse token budgets. Consequently, our method proves efficacy in maintaining high performance with different VLMs.

### 4.3 Validating Generality for Autonomous Driving Scene Understanding and Planning

To assess the generalization abilities of our pruning strategy, we conduct a comparative study on scene understanding and driving planning tasks. This evaluation utilized the Senna model [10] and its

Table 1: Comparison of our methods with other training-free token pruning methods. "Avg.tokens" refers to the average number of tokens that will be retained. Ratio represents the average percentage of performance maintained at the corresponding reduction ratio.

| Method | Present at | Avg. tokens | MME | MMB | SQA | GQA | TextVQA | Ratio | FLOPs |
|---|---|---|---|---|---|---|---|---|---|
| LLaVA-1.5-7B | NeurIPS'24 | 576 | 1862 | 64.7 | 69.5 | 61.9 | 58.2 | 100% | 100% |
| ToMe [15] | arXiv'22 | 192 | 1563 | 60.5 | 65.2 | 54.3 | 52.1 | 89.9% | 44.3% |
| FastV [18] | ECCV'24 | 192 | 1612 | 61.2 | 67.3 | 52.7 | 52.5 | 90.6% | 45.7% |
| SparseVLM [20] | arXiv'24 | 192 | 1721 | 62.5 | 69.1 | 57.6 | 56.3 | 95.5% | 46.3% |
| PDrop [13] | CVPR'25 | 192 | 1797 | 63.3 | 69.2 | 57.3 | 56.5 | 96.8% | 43.9% |
| **Ours** | - | 192 | **1832** | **64.9** | **69.6** | **60.4** | **57.7** | **99.0%** | **42.9%** |
| ToMe [15] | arXiv'22 | 128 | 1343 | 53.3 | 59.6 | 52.4 | 49.1 | 81.1% | 35.1% |
| FastV [18] | ECCV'24 | 128 | 1490 | 56.1 | 60.2 | 49.6 | 50.6 | 83.9% | 36.8% |
| SparseVLM [20] | arXiv'24 | 128 | 1696 | 60.0 | 67.1 | 56.0 | 54.9 | 93.0% | 37.3% |
| PDrop [13] | CVPR'25 | 128 | 1761 | 61.6 | 68.4 | 57.1 | 56.6 | 95.6% | 35.1% |
| **Ours** | - | 128 | **1785** | **64.3** | **69.7** | **59.9** | **57.4** | **98.1%** | **33.7%** |
| ToMe [15] | arXiv'22 | 64 | 1138 | 43.7 | 50.0 | 48.6 | 45.3 | 70.5% | 25.7% |
| FastV [18] | ECCV'24 | 64 | 1256 | 48.0 | 51.1 | 46.1 | 47.8 | 73.7% | 27.9% |
| SparseVLM [20] | arXiv'24 | 64 | 1505 | 56.2 | 62.2 | 52.7 | 51.8 | 85.9% | 28.2% |
| PDrop [13] | CVPR'25 | 64 | 1561 | 58.8 | 69.0 | 47.5 | 50.6 | 87.6% | 25.5% |
| **Ours** | - | 64 | **1745** | **63.6** | **69.6** | **57.7** | **57.1** | **96.7%** | **23.2%** |

Table 2: Comparison of different pruning methods on LLaVA-NeXT-7B. Performance data for the compared methods are drawn from prior publications. For methods where results on LLaVA-NeXT were not provided in existing literature, we have reproduced their experiments and present a comparative analysis against our approach in the supplementary materials.

| Method | Present at | Tokens | VQA$^{V2}$ | GQA | TextVQA | POPE | MME | Ratio |
|---|---|---|---|---|---|---|---|---|
| LLAVA-NeXT-7B | NeurIPS'24 | 2880 | 81.2 | 62.9 | 59.6 | 86.3 | 1513.8 | 100.0% |
| FastV [18] | ECCV'24 | 640 | 78.9 | 60.4 | 58.4 | 83.1 | 1477.3 | 97.0% |
| SparseVLM [20] | arXiv'24 | 640 | 78.2 | 59.1 | 56.2 | 80.9 | 1456.3 | 94.9% |
| VisionZip [17] | CVPR'25 | 640 | 79.2 | 60.1 | 58.5 | 82.2 | 1468.4 | 96.7% |
| FasterVLM [51] | arXiv'24 | 640 | 79.8 | 61.6 | 59.3 | 85.9 | 1480.7 | 98.6% |
| **Ours** | – | 640 | **80.5** | **62.6** | **59.6** | **86.7** | **1515.7** | **99.7%** |
| FastV [18] | ECCV'24 | 320 | 71.9 | 55.9 | 55.7 | 71.7 | 1282.9 | 87.7% |
| SparseVLM [20] | arXiv'24 | 320 | 71.4 | 56.5 | 52.4 | 73.5 | 1342.7 | 87.9% |
| VisionZip [17] | CVPR'25 | 320 | 74.2 | 58.1 | 55.3 | 75.0 | 1348.8 | 90.5% |
| FasterVLM [51] | arXiv'24 | 320 | 75.7 | 58.4 | 57.6 | 80.4 | 1370.1 | 93.3% |
| **Ours** | – | 320 | **78.9** | **61.3** | **59.5** | **85.6** | **1471.6** | **98.2%** |
| FastV [18] | ECCV'24 | 160 | 61.8 | 49.8 | 51.9 | 51.7 | 1079.5 | 74.7% |
| SparseVLM [20] | arXiv'24 | 160 | 62.2 | 50.2 | 45.1 | 54.6 | 1167.1 | 74.9% |
| VisionZip [17] | CVPR'25 | 160 | 67.3 | 54.3 | 54.7 | 59.4 | 1239.7 | 82.3% |
| FasterVLM [51] | arXiv'24 | 160 | 70.6 | 54.7 | 56.0 | 72.9 | 1226.0 | 86.7% |
| **Ours** | – | 160 | **76.4** | **59.4** | **57.2** | **81.4** | **1457.0** | **94.9%** |

associated customized nuScenes dataset. The official task adopts "Planning Accuracy" as the official evaluation metric. Tab. 3 reports performance retention when applying different pruning techniques to the Senna VLA model [10]. *Notably, our method is applied to autonomous driving without any hyper-parameter tuning.* As detailed in Tab. 3, our method consistently surpass all competing methods across these diverse pruning ratios. For instance, at a 25% token retention level, our approach achieved a remarkable 111.23% relative accuracy, outperforming not only PyramidDrop at 98.89% but also the original unpruned model. These findings strongly suggest that our pruning strategy effectively preserves essential visual information even within challenging, large-scale real-world scenes. An intriguing observation emerged as our pruned model occasionally outperformed the full model, indicating the potential presence of detrimental noisy visual tokens in VLAs trained on sparse, large-scale datasets. We intend to investigate this compelling finding in future research.

Table 3: The performance retention ratio on the nuScenes scene understanding and planning tasks. Language-based data and baseline model come from Senna [10].

| Retention | 26/128 (20%) | 32/128 (25%) | 38/128 (30%) | 45/128 (35%) | 51/128 (40%) |
|---|---|---|---|---|---|
| FasterVLM [51] | 55.05% | 52.81% | 47.19% | 45.50% | 49.43% |
| SparseVLM [20] | 84.29% | 93.83% | 95.52% | 97.76% | 101.13% |
| PyramidDrop [13] | 94.94% | 98.89% | 96.07% | 98.29% | 100.55% |
| **Ours** | **96.63%** | **111.23%** | **106.75%** | **105.06%** | **104.51%** |

Table 4: Impact of different indicators for complexity and pruning schedules.

| **Metric** | TextVQA | | GQA | |
|---|---|---|---|---|
| | 64 | 128 | 64 | 128 |
| Static Logistic | 55.1 | 55.9 | 55.6 | 57.6 |
| Cosine Similarity | 55.8 | 56.7 | 56.1 | 57.9 |
| Average Attention | 56.2 | 56.9 | 56.7 | 58.4 |
| **Mutual information** | **57.1** | **57.4** | **57.7** | **59.9** |

(a) Different indicators for complexity

| **Curve** | TextVQA | | GQA | |
|---|---|---|---|---|
| | 64 | 128 | 64 | 128 |
| Linear | 54.1 | 55.5 | 52.7 | 54.9 |
| Tanh | 56.6 | 56.9 | 55.2 | 57.5 |
| Exponential | 56.1 | 56.6 | 54.2 | 56.3 |
| **Logistic** | **57.1** | **57.4** | **57.7** | **59.9** |

(b) Different pruning schedule curves

## 4.4 Ablation Studies

**Different Visual–Textual Relationship Indicators.** We evaluate three distinct indicators to quantify the strength of visual-textual correlation, including our proposed mutual information (MI), the average cross-attention magnitude, and the cosine similarity between visual and text embeddings. The empirical results presented in Tab. 4(a) reveal that our mutual information approach, which draws inspiration from cognitive science, achieves better average accuracy. This outcome substantiates the superior effectiveness of mutual information in guiding token retention strategies.

**Impact of Pruning Schedule Curve.** To specifically isolate the impact of different scheduling functions, we incorporated four canonical curves namely linear, logistic (sigmoid), hyperbolic tangent (tanh), and exponential into our AutoPrune framework. We then evaluate their respective performances on the GQA [49] and TextVQA [50] datasets. The results, presented in Tab. 4(b), indicate that the logistic curve yields superior performance compared to the other scheduling functions. This finding further corroborates the previously discussed cognitive science principles, underscoring the efficacy of the logistic function in modeling attentional allocation.

Due to space limitations, additional analysis including 1) comparison with more methods, 2) on other benchmarks, and 3) adaptations with flash-attention, are presented in the supplementary materials.

## 5 Conclusion and Limitation

In conclusion, this paper address the computational burden of long visual sequences in VLMs by introducing Complexity-Adaptive Pruning (AutoPrune), a novel training-free framework. Inspired by cognitive neuroscience, AutoPrune quantifies sample and task complexity via mutual information between early visual and textual tokens, mapping this to individualized, budget-constrained logistic retention curves that dictate token pruning across decoder layers. Our extensive evaluations demonstrate that AutoPrune offers a simple, generalizable, and highly effective solution for enabling efficient real-time multimodal reasoning and embodied intelligence. Our investigation reveals a nuance in attention distribution, a limitation also observed in related studies. While token importance generally decreases with decoder depth, our findings (Fig. 1) show deeper layers can occasionally retain more critical tokens than shallower ones. Although our work advances depth-aware pruning, further refinement is needed for strategies to dynamically match this variable importance distribution across network depth. We leave this in future research.

## Acknowledgements

This work was supported in part by the Natural Science Foundation of China (Grant No. 62036011, 62422317, U22B2056, 62503323), the Beijing Natural Science Foundation (Grant No. JQ22014, L223003, JQ24022). The work of Yufan Liu and Weiming Hu was also supported in part by the Natural Science Foundation of China (Grant No. 62192782, 62503323, U2441241, 62372451, 62192785, 62372082), the CAAI-Ant Group Research Fund (Grant No. CAAI-MYJJ 2024-02), the Young Elite Scientists Sponsorship Program by CAST (Grant No. 2024QNRC001).

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
