# OpenReview forum: "Each Complexity Deserves a Pruning Policy"
_NeurIPS.cc/2025/Conference — NeurIPS 2025 poster_

### Official Review · Reviewer_zvmf · 2025-06-04

**Clarity:** 1
**Significance:** 3
**Originality:** 2
**Rating:** 4
**Confidence:** 4

**Summary:**

AdaPrune is a method for flexibly removing visual tokens in large vision–language models (LVLMs) based on input complexity. Unlike prior approaches that apply a fixed, layer-wise token removal schedule uniformly across all samples and tasks, AdaPrune measures the mutual information between visual and textual tokens and generates a logistic retention curve tailored to each sample’s complexity. This ensures that only the necessary tokens are retained under a given computational budget, drastically reducing computation.

**Questions:**

Refer to Weaknesses

**Ethical Concerns:**

["NO or VERY MINOR ethics concerns only"]

**Final Justification:**

The authors addressed the concerns that initially I mentioned, so I've updated the score. Especially, I recommend the authors should handle the motivation part to be clarified and more provide examples of figures regarding dataset-wise retention ratio curves that is sure to be interesting.

**Limitations:**

yes

**Quality:**

2

**Strengths And Weaknesses:**

**Strength**

- Training-Free Framework: Offers a plug-and-play solution that requires no additional training.
- Cognitively Inspired: Leverages visual–textual mutual information to quantify complexity, mimicking biological attention mechanisms.
- Per-Sample Token Control: Employs individualized, budget-constrained logistic retention curves to dynamically prune tokens per decoder layer.

---

**Weaknesses**

- Unclear writing: It makes the reader get easily confused with the terminology and the motivation of neuroscience. This writing has so many expert-level terminology for brain and redendunt detail explanation. These easily made me distract from the main point of what the authors want to say. Line 53-63 and Line 154-165 are most paragraphs that I can't understand much. Even, the rest of the writings is also somewhat hard to read smoothly. Unless really focusing on the writing, it is easy to miss some parts. I don't think the writing with hard and complex terminology is great paper at least for AI community due to rapid change and development. Please make the writing narrowed for the scope of vision language-perspective and pruning-perspective, although the authors want to say everything that you can come up with the expert concept.

- Vague standard for simple and complex: As in figure 1, it represents layer-wise visual text interaction patterns where simeple task (object identification) and complex task (reasoning-intensive) feels some awkward. This is because according to question, I can recognize whether it is simple or comlpex task. However, in the perspective of feature visualization and saliency map, i don't think it converges on the salient region and remains stable, even when simple task is given. I think the problem setup should be more solid or more explored.

- Applicability to more recent released models: Your models feel so out-dated because recently, there are so many VLMs such as Qwen2/2.5-VL or InternVL2.5/3-VL. Because none of these recent models, it may limits the evidecneof your solid experiments and applicability.

- Lack of interesting phenomenon: Figure2 seems so interesting results but there are none of additional results for other benchmarks, because Figures2 is achieved on TextVQA only. I would highly recommend that kinda graphs for numerous benchmarks and have some analysis for them.

- Challenging Evaluation Benchmarks: Your selected evaluation benchmarks are mainly from easy benchmarks including just selecting the multiplie choice or simple answer. However, there are so many challenging benchmarks including free-form answer (long context answer) like MM-Vet (or -v2) and really challening knowledge evlauation benchmark such as MMMU.

- Overall, I think the motivation is somewhat lack and there are some redundent explanation beyond the main point that the authors should convey. This paper could be improved in terms of writing quality and clear standard to discriminate how we can acquire insight from salieny map because it is merely obvious phenomenon that many papers have reported. Furthermore, this paper must be more solid and having dense experimental resulsts for consistent phenomenon of numerous benchmarks and applicability to recent models.

---

> ### Author Rebuttal · Authors · 2025-07-31
>
> Thank you for your detailed review and thoughtful comments. We appreciate your emphasis on our training‑free plug‑and‑play design, cognitively inspired MI, and per‑sample budgeted logistic pruning.
>
> **Q1: The introduction of neuroscience.**
>
> **A1:** Thank you for the insightful comment. We will revise the manuscript to simplify our prose and reduce the use of specialized neuroscience terminology. We included those details because the human brain is intrinsically skilled at reasoning and modern vision–language models demonstrate similar inference behaviors. Many approaches draw inspiration from human reasoning processes and have achieved excellent results. For example, chain‑of‑thought prompting simulates step‑by‑step human reasoning.  Moreover, dual‑system, brain‑inspired approaches, known as fast and slow thinking, have been widely applied in autonomous driving and embodied intelligence domains [1–3]. Our approach aims to leverage these human visual reasoning strategies to enhance model performance. Without neuroscience‑inspired heuristics, designing an effective token pruning mechanism would be very challenging.
>
> [1] Fast-in-Slow: A Dual-System Foundation Model Unifying Fast Manipulation within Slow Reasoning
>
> [2] Agentic Robot: A Brain-Inspired Framework for Vision-Language-Action Models in Embodied Agents
>
> [3] Language‑Conditioned Robotic Manipulation with Fast and Slow Thinking
>
> **Q2: “Simple vs. Complex” in Fig. 1.**
>
> **A2:** Thank you for raising this point. We apologize for the confusion. Fig. 1 is intended to illustrate two observations: (1) a shared depth‑wise trend across tasks, and (2) different convergence dynamics between simple and complex questions.
>
> 1. **Shared trend across tasks.** As depth increases, the visual modality receives progressively less attention, as shown in Fig. 1, by Layer 31 the model’s focus almost completely shifts toward language tokens. Since this depth‑wise reduction of visual attention has been widely discussed in prior work, we did not emphasize it in the main text; we will make this point explicit to avoid misinterpretation.
> 2. **Different** **convergence** **dynamics by task difficulty.** For simple queries (Q1–Q3, object identification), attention shifts rapidly toward the target between layers 2 and 16, often by layer 4 or layer 8, and then remains stable. In contrast, for complex, reasoning-intensive queries (Q4–Q6), the model does not correctly localize the object by layer 4 and only begins to identify and stabilize the relevant region between layers 8 and 16. These divergent layer‑wise interaction patterns show that simple and complex questions follow distinct processing trajectories. This observation inspired our layer and depth aware pruning framework, which dynamically adjusts token retention to each task’s complexity and attention dynamics.
> 3. **On saliency stability.** Your observation that saliency “does not converge on the most salient region” at large depth is consistent with (1) above: late layers are increasingly language‑dominated. Our analysis focuses on the emergence and stabilization of correct visual grounding in intermediate layers, where the simple vs. complex distinction is most diagnostic.
>
> **Q3: Applicability to more recently released models.**
>
> **A3:** We thank the reviewer for this valuable suggestion. As shown in Tab. 2 of the manuscript, we have already validated our approach on LLaVA‑NeXT, which was released contemporaneously with Qwen2. In response, we have extended our evaluation to include Qwen2‑VL. As shown below, our method continues to deliver consistent improvements over the prior state of the art on this model, demonstrating that it generalizes robustly across the latest vision–language models.
>
> | MMB               | Token   | MMB↑     | POPE↑    | TextVQA↑ |
> | ----------------- | ------- | -------- | -------- | -------- |
> | Qwen-2VL          | Dynamic | 80.5     | 86.4     | 84.3     |
> | SparseVLM (ICML 25) | 600     | 79.6     | 86.5     | 80.3     |
> | **Ours**          | **600** | **80.1** | **86.5** | **81.7** |
> | SparseVLM (ICML 25) | 400     | 79.0     | 85.8     | 77.1     |
> | **Ours**          | **400** | **79.4** | **86.1** | **78.6** |
>
> **Q4: Visualizations for numerous benchmarks.**
>
> **A4:** Because mutual‑information values inherently occupy a similar numerical range across datasets, the resulting curves exhibit only minor visual differences. Unfortunately, the NeurIPS rebuttal format does not permit the inclusion of new figures. We will, however, incorporate these additional plots and accompanying analysis for multiple benchmarks in the revision.
>
> **Q5: More challenging benchmarks.**
>
> **A5:** To address this, we have added experiments on the MMMU benchmark. As shown below, our approach continues to outperform the prior state of the art, confirming that it generalizes robustly even on more challenging, free‑form and knowledge‑heavy datasets. The full LLaVA-1.5-7B achieves an accuracy of 34.9 %. Even when retaining only 75 %, 50 % and 11.1 % of the tokens our method still surpasses the competing techniques. These findings demonstrate the strong generalizability of our approach.
>
> | Method               | 75%      | 50%      | 11.1%    |
> | -------------------- | -------- | -------- | -------- |
> | FasterVLM（ICCV 25） | 34.1     | 33.9     | 33.7     |
> | PyDrop (CVPR 25)     | 33.6     | 33.4     | 32.2     |
> | **Ours**             | **34.4** | **34.2** | **34.3** |
>
> **Q6: Motivation of AdaPrune.**
>
> **A6:** We explain our motivation from the following two aspects:
>
> - **Limitations of existing approaches.** Although previous pruning methods typically rely on the well‑known observation that visual attention mass decreases with layer depth, they apply the same pruning schedule to all tasks. This approach fails to adapt to each question’s unique visual–textual requirements and to the specific locations where visual evidence appears.
> - **Our perspective.** We complement attention‑mass views with spatial dynamics of attention: how and when the model localizes relevant regions as depth increases. Inspired by human problem solving (rapid fixation for simple cases vs. search for complex ones) and supported by our visualizations, we show that simple queries stabilize early, whereas reasoning‑intensive queries exhibit delayed or shifting localization. This depth‑wise localization dynamic directly motivates a task‑adaptive pruning policy rather than a one‑size‑fits‑all schedule.
>
> **Q7: Insight from** **the** **salient map.**
>
> **A7:** Prior work (e.g., FastV) predominantly reports that attention magnitude on visual tokens shrinks with depth. However, in VLM pruning the focus of attention and its evolution during reasoning have received little scrutiny. Our saliency maps make this spatiotemporal relocation explicit across different question types. We then model these regularities to guide pruning decisions: stable regions trigger more aggressive pruning elsewhere, whereas shifting patterns call for conservative pruning until convergence. In the revised manuscript, we will include a clear operational definition of our saliency metric and specify the criteria that render it actionable.

---

> ### Comment · Reviewer_zvmf · 2025-07-31
> **Response**
>
> I've carefully read the rebuttals and thank you for your efforts. The concerns that I was posing initially are mostly addressed. I've updated the scores towards leaning accept. I suggest that authors reflect on not only my comments but also others' ones, and please make sure the clarification on the manuscript as well.

---

> > ### Author Response · Authors · 2025-08-01
> > **Thank Reviewer zvmf for the increased score**
> >
> > We are grateful for your “leaning accept” assessment. We will incorporate the additional results and technical clarifications you suggested and ensure the revision clearly reflects your comments as well as those of the other reviewers.

---

### Official Review · Reviewer_qWyB · 2025-06-28

**Clarity:** 2
**Significance:** 3
**Originality:** 3
**Rating:** 5
**Confidence:** 3

**Summary:**

This paper introduces AdaPrune, a training-free token pruning framework that, drawing inspiration from neuroscience, adaptively selects layer-wise token retention ratios subject to a global constraint on inference compute. The neuroscience background suggests that humans typically narrow their focus to constrained locations in visual scenes in early processing stages when the task is simple, while they conversely preserve several hypotheses for longer when the task is complex. This work shows an analogous pattern in Vision-Language Models (VLMs), where the attention maps may "converge" in early or late layers, depending on the textual query associated with it.

Given this empirical evidence, AdaPrune maps a notion of query difficulty (computed via mutual information between visual and textual tokens) to a logistic curve that determines how many tokens to retain at each layer. Specifically, the logistic curve maps a layer index to a retention ratio, and the slope and inflection point of the curve are linear functions of the aforementioned mutual information. Such a curve is further normalised to ensure its integral (which reflects the total number of tokens processed end-to-end during inference) matches a given compute constraint.

Successful experiments are conducted on standard Vision-Language benchmarks with LLaVA-v1.5-7B and LLaVA-Next-7B, as well as with Senna, a Vision-Language-Action model employed for autonomous driving.

**Questions:**

For a productive rebuttal, I would probably start with the following questions (expanding on the weaknesses above).

1. Is Mutual Information for a single visual token the criterion used to choose which tokens are retained after determining a retention ratio with the logistic curve? So, something like Eq. (3) of the paper, where the outer summation over visual tokens is unpacked into $N_v$ scalars determining the importance scores of each of the visual tokens? I have reviewed Section D of the supplementary materials, where an ad-hoc importance score is defined for the first layer. Line 40 therein states: *"We assign each visual token an importance score based on the textual-visual attention weight."* It is, however, unclear to me if this holds for all layers or only for the first one. In general, could the authors please clarify this aspect about token selection?

2. Could the authors please clarify how the parameters of the logistic curve are selected, i.e., which data was used, how many data points, etc?

3. In my understanding, the logistic curve computed from Mutual Information determines how tokens are progressively dropped in the stacked blocks of the LLM decoder, meeting the global compute constraint after normalisation, but at which layer is this Mutual Information computed? While reading, I assumed this computation happens immediately in the first block, and the derived scalar determines computation throughout the rest of the decoder, but could the authors please clarify this aspect as well?

**Ethical Concerns:**

["NO or VERY MINOR ethics concerns only"]

**Final Justification:**

My review initially expressed concerns about the lack of details for three aspects:
- Scoring function used for token selection;
- Choice of the layer for MI estimation;
- Hyperparameter configurations;

The authors explained the details with care during the initial phases of the discussion, as well as stated they will clarify them in the revised manuscript. Therefore, I am increasing my score.

**Limitations:**

The final section of the paper is titled "Conclusions and Limitations", but it looks more like a "Conclusions" Section rather than a "Limitations" one. However, I do not see important unaddressed considerations.

**Paper Formatting Concerns:**

I don't see any formatting concerns.

**Quality:**

3

**Strengths And Weaknesses:**

**Strengths**.
1. The paper is well-written and nicely organised;
2. The underlying idea is simple, which I believe is a plus, is carefully designed (I particularly liked the overall reframing of attention scores into MI) and has a sound motivation;
3. Experiments are comprehensive, and extending towards VLAs with fixed hyperparameters shows the practical applicability of the proposed framework beyond academic benchmarks;


**Weaknesses**.
I think the major weakness of this paper is that some passages of the methodological section are unclear or partially omitted. For example:
1. In my understanding, the adaptive logistic curve provides a per-layer token retention ratio based on the overall scalar-valued mutual information, but I see no detail about which tokens are retained and which are discarded;
2. Other than the estimated mutual information, the logistic curve depends on some parameters ($k_0$, $\gamma$, $x_o$, $\beta$) which should be defined a priori. It's probably amiss for this paper not to report how these are selected;

---

> ### Author Rebuttal · Authors · 2025-07-31
>
> Thank you for your detailed review and thoughtful comments. We’re grateful for your comments on the paper’s clarity and organization, the simple yet well‑motivated reframing of attention as MI, and the comprehensive experiments.
>
> **Q1: Details about which tokens are retained.**
>
> **A1:** We thank the reviewer for requesting more detail on token retention.
>
> - **First layer.** Within the initial processing layer, the fusion of textual and visual modalities remains insufficient. As shown in Sec. D of the supplementary materials, we use Cross‑Modal Weighted Pruning. We compute each visual token’s importance score by combining its self‑attention weight with its textual attention weight. This hybrid scoring prevents discarding tokens that carry critical visual content before stronger cross‑modal interactions form.
> - **Subsequent layers.** Visual tokens are scored by cross‑modal attention between image and text features, and we select the top $k$ tokens for downstream processing, with $k$ governed by our dynamic retention schedule. Our study targets dynamic control of the pruning budget rather than the design of ranking heuristics. To keep the analysis simple and comparable, we therefore use raw cross‑modal attention weights as the ranking criterion in all experiments. More elaborate scoring strategies reported in prior work could yield additional gains. This direction is orthogonal to our contribution and is left for future research.
>
> **Q2: Selection of** $(k_0, \gamma, x_0, \beta)$**.**
>
> **A3:** Thank you for raising this important concern. Below we provide additional detail on our hyperparameter choices and token‑selection criteria:
>
> 1. **Hyperparameter** **mapping**
>
>    The mapping from mutual‑information score $I_q(V, T)$ to the slope $k_q$ and the inflection point $x_0^q$ is defined by
>
>    $$k_q = k_0 - \gamma\,I_q(V, T),\quad  x_0^q = x_0 + \beta\,I_q(V, T). $$
>
>    To demonstrate robustness, we fix these values across all experiments (See below for parameter selection). Specifically we set
>
>    $$k_0 = 0.4,\quad \gamma = 0.3,\quad x_0 = 15,\quad \beta = 0.3. $$
>
>    We apply only one clamp, enforcing $k_q \ge 0$ so that the selection process can only decrease the token number and never increase it. We do not apply any calibration to the mutual‑information values.
>
> 2. **Token‑selection criteria**
>
>    Visual tokens are scored by cross‑modal attention between image and text features, and we select the top $k$ tokens for downstream processing, with $k$ governed by our dynamic retention schedule. Our study targets dynamic control of the pruning budget rather than the design of ranking heuristics. To keep the analysis simple and comparable, we therefore use raw cross‑modal attention weights as the ranking criterion in all experiments. Although more sophisticated scoring mechanisms described in prior work may yield further gains, such enhancements fall outside the scope of the present study and are reserved for future investigation.
>
> 3. **Robustness across hyperparameters**
>
>    - **Parameter selection strategy. 1) Curve-based criteria.** As shown in Fig. 2, the retention curve must clearly distinguish easy from hard samples. We therefore choose parameters that provide sufficient dispersion, avoiding curves that concentrate around the midpoint or flatten at the extremes, so that the pruning schedule remains balanced. **2) Empirical** **validation****.** A small series of preliminary experiments yielded the following hyperparameters without extensive tuning:
>
>      $$k_0 = 0.4,\quad \gamma = 0.3,\quad x_0 = 15,\quad \beta = 0.3. $$
>
>      Specifying each value to one decimal place demonstrates that no overfitting occurred on any particular dataset.
>
>    - **Supplementary experiments.** To validate the robustness of these settings, we ran additional experiments under alternative hyperparameter configurations. Although some variants produced slightly improved results (see the table below), we elected to retain the original fixed values in order to ensure fairness and consistency across all datasets.
>
>      | Configuration                                      | TextVQA(token=64)↑ | TextVQA(token=128)↑ | GQA(token=64)↑ | GQA(token=128)↑ |
>      | -------------------------------------------------- | ------------------ | ------------------- | -------------- | --------------- |
>      | $k_0=0.3, \gamma=0.2, x_0=16, \beta=0.5$           | 57.2               | 57.6                | 57.3           | 59.4            |
>      | $k_0=0.5, \gamma=0.4, x_0=14, \beta=0.2 $          | 56.9               | 57.3                | 57.6           | 59.6            |
>      | $k_0=0.4, \gamma=0.3, x_0=15, \beta=0.3 $(default) | 57.1               | 57.4                | 57.7           | 59.9            |
>
> **Q3: Which layer is this Mutual Information computed?**
>
> **A3:**  We appreciate the reviewer's valuable suggestion.
>
> 1. **Optimal Layer Selection for MI Estimation.** Thank you for raising this important and insightful concern. Estimating mutual information too early can indeed introduce inaccuracies, which is why we perform MI estimation beginning at the second layer of the network. To address your question, we have conducted additional experiments measuring MI at various layers. The results, summarized in the table below, reveal two key trends. First, estimating MI too early (at layer 1) injects spurious signals into the pruning schedule, causing the model’s accuracy to suffer. Second, delaying MI estimation until very late layers front‑loads an excessive share of the fixed compute budget, which also diminishes overall performance. Estimating MI at layer 2 therefore achieves the optimal balance between reliable signal extraction and efficient budget utilization. Notably, in layers where MI is not estimated, we employ a linear pruning schedule in our experiments.
>
>    | **Layer** | TextVQA(token=64)↑ | TextVQA(token=128)↑ | GQA(token=64)↑ | GQA(token=128)↑ |
>    | --------- | ------------------ | ------------------- | -------------- | --------------- |
>    | **1**     | 56.4               | 56.8                | 56.5           | 59.1            |
>    | **2**     | 57.1               | 57.4                | 57.7           | 59.9            |
>    | **3**     | 56.9               | 57.2                | 57.4           | 59.5            |
>    | **4**     | 56.6               | 57.0                | 57.2           | 59.4            |
>
> 2. **Cross‑Modal Hybrid Token Scoring Mechanism.** Within the first layer, the fusion of textual and visual modalities remains insufficient. As shown in Sec. D of the supplementary materials, we use Cross‑Modal Weighted Pruning. We compute each visual token’s importance score by combining its self‑attention weight with its textual attention weight. This hybrid scoring prevents discarding tokens that carry critical visual content before stronger cross‑modal interactions form.
>
> **Q4: Limitations: Important unaddressed considerations.**
>
> **A4:** We agree that our “Conclusions and Limitations” section can be strengthened by more explicitly acknowledging key shortcomings. We have identified and will add the following limitations to the revised manuscript:
>
> 1. **Lack of a token recall mechanism.** Although token importance generally decreases with decoder depth, our analysis in Fig. 1 reveals that deeper layers sometimes retain tokens that are more critical than those in shallower layers. This finding motivates the development of a mechanism to recover valuable tokens that were discarded in early layers.
> 2. **Application to video-based VLMs.** Our pruning method is, in principle, directly transferable to video vision-language models, but we have not yet introduced any temporal modeling. Incorporating temporal information into our token pruning framework represents a meaningful direction for future research.
>
> We will incorporate these points into the Limitations section in the revision.

---

> > ### Comment · Reviewer_qWyB · 2025-08-01
> > **Thank you! Let's Discuss**
> >
> > Dear Authors,
> > Thank you for the detailed response! Let me reply according to the main points:
> >
> > **Criterion for token selection.** Just for me to be extra sure: the sentence *"visual tokens are scored by cross‑modal attention between image and text features"* means that the score $s_v$ assigned to visual token $v$ is
> >
> > $s_v = \sum_{t \in T} \alpha_{t,v}$
> >
> > where $\alpha_{t,v}$ is the softmax-normalized causal attention score from textual token $t$ to visual token $v$, and $T$ is the set of all textual tokens. Is my understanding correct? I definitely agree with you that the contribution of this paper is the *"dynamic control of the pruning budget rather than the design of ranking heuristics"* and that a scoring function for tokens is orthogonal to this work. However, I believe token selection is overall an important detail not to overlook in a pruning paper, and ideally should be clear from the main body, so that alone is sufficient for anyone to reproduce the work. Would you be open to moving this aspect somewhere in the Methodological Section?
> >
> > **Layer choice for MI estimation.** Thank you for this analysis! Did you perhaps experiment with models where the overall number of LLM decoder blocks is significantly different? I wonder whether the absolute layer 2 index would generalize better if expressed in terms of fractional compute spent within the LLM, e.g., as a simple example, using Layer-2 if the LLM has, say, 18 layers, while using Layer-4 if the LLM has 36 layers. (Note: I think the response is sufficient, no need to perform this additional experiment).
> >
> > **Hyperparams for logistic curve.** Thank you for conducting additional experiments with different hyperparameter configs. I think stability *w.r.t.* hyperparameter choice is indeed a critical aspect for a pruning scheme, and I appreciate that this work demonstrates it. For completeness, could you please clarify what you mean by *"A small series of preliminary experiments yielded the following hyperparameters without extensive tuning"*? On which data (or benchmarks) did you run these experiments, how the ranges for grid search were set, and so on?
> >
> >
> > All the best,\
> > Reviewer `qWyB`

---

> > > ### Author Response · Authors · 2025-08-01
> > > **Thank Reviewer qWyB for the reply**
> > >
> > > **Q1: Criterion for token selection.**
> > >
> > > **A1:** Thank you for articulating that so clearly.
> > >
> > > 1. **Details of the token selection.** Your interpretation is correct. We score each visual token by aggregating the softmax-normalized causal attention weights received from all textual tokens, and use that aggregate as its importance measure.
> > > 2. **Making token selection explicit in the Method section.** We fully agree that token selection is a substantive detail for pruning work and should not remain implicit. To improve clarity and reproducibility, we will add a precise and concise description of this scoring mechanism to the Method section.
> > >
> > > **Q2: Layer choice for MI estimation.**
> > >
> > > **A2:** We thank you for this insightful and valuable suggestion. So far we have evaluated models with only moderate differences in decoder depth, specifically LLaVA-7B and LLaVA-13B whose language backbones have 32 and 40 decoder layers. In both cases estimating mutual information at layer 2 yields the best pruning results. We have not yet examined cases with much larger depth gaps, such as 64 layers or more. We agree that using relative or fractional layer indices could potentially benefit generalization with varying depths, but this remains a hypothesis that requires further investigation. We will leave this to future work.
> > >
> > > **Q3: Hyperparameters for logistic curve.**
> > >
> > > **A3:** Thank you for the thoughtful question. We initialized the logistic-curve hyperparameters on GQA with $k_0 = 0.5, \gamma = 0.5, x_0 = 15,\beta = 0.5$. To avoid overfitting, we conducted a coarse search over a small set of concrete settings rather than an exhaustive grid. Specifically, we evaluated 5 configurations for the pair $(k_0, \gamma)$ by stepping 0.1 around the seed value 0.5, and 4 configurations for the pair $(x_0, \beta)$ by stepping 1 and 0.1 around 15 and 0.5. We selected the configuration that produced the most stable behavior across different pruning ratios. A finer and more exhaustive grid search might further improve tuning, but it would also increase the risk of overfitting and would not align with our goal of keeping the procedure simple and broadly generalizable.

---

> > > > ### Comment · Reviewer_qWyB · 2025-08-01
> > > >
> > > > Dear Authors,
> > > >
> > > > Thank you for the prompt response. My doubts were cleared, and I am therefore increasing my score.
> > > >
> > > > All the best,\
> > > > Reviewer `qWyB`

---

> > > > > ### Author Response · Authors · 2025-08-02
> > > > > **Thank Reviewer qWyB for the increased score**
> > > > >
> > > > > We are grateful for your increased score. We will incorporate the additional results and technical details following your suggestions.

---

### Official Review · Reviewer_u9ow · 2025-06-30

**Clarity:** 2
**Significance:** 2
**Originality:** 3
**Rating:** 4
**Confidence:** 3

**Summary:**

This paper concentrates on the vLLM's inference acceleration. In particular, previous methods typically employ heuristics layer-specific pruning strategies, and the overall pruning schedule is fixed and applied uniformly to all input samples and tasks, regardless of the samples, tasks or layers. In this work, inspired by the cognitive science, the authors introduce Complexity-Adaptive Pruning (AdaPrune), which is a training-free, plug-and-play framework that tailors pruning policies to varying sample and task complexities. The method is based the mutual information between visual and text inputs, and quantifies the compexity of a sample. Then, the pruning policy is designed based on the mutual information. Experiments are done on LLaVA and VQA benchmarks.

**Questions:**

See above

**Ethical Concerns:**

["NO or VERY MINOR ethics concerns only"]

**Final Justification:**

Thanks for the response, my concerns are solved, I have raised my score. Please add these clarification and experiments in the final version.

**Limitations:**

See above

**Quality:**

3

**Strengths And Weaknesses:**

The strengths are as follows:
1.This work is well motivated. Recent training free acceleration methods mainly rely on hand-crafted metrics and the pruing policy is fixed regardless of the layers. This paper proposes a sample or task complexity adaptied pruing model.
2.The idea based on cognitive science and mutual information is interesting. Using mutual information between visual and text input as a proxy objective to measure the sample's comlexity and prune tokens is new to me. It sounds reasonable.

The weakness, also my questions are as follows:
1.Why the Budget-Constrained Logistic Retention is designed as Eq.5. It is not easy to understand how this formulation is introduced and why it is good to achieve the target?
2.In Figure 2, Why simple samples mainly lie in retention rate between 0.2-0.4 in both low and high layers? For hard samples, it is easy to understand the retention rate is high in low layers, but how to understand the retention rate is low in high layers? Do we need more information for hard samples in both low and high layers?
3.The experiments could be improved.
a).The test set of VQA task is very simple, only Yes or No, or answer selection. Are there any other more complex generation task for evaluation? Image caption or image generaion?
b).Some related works are not included and discussed. Recently, there are some other training-free pruning methods, such as VTW[1], Turbo[2] and Folder[3]. These methods should be compared and discussed.
c).In Table 3, the reported performance is  a relative improvement. What is the absolute performance of compared baseline model, it is worthwhile to take the absolute performance into cosideration to check the relative improvement.

[1]Boosting Multimodal Large Language Models with Visual Tokens Withdrawal for Rapid Inference AAAI'25
[2]Turbo: Informativity-Driven Acceleration Plug-In for Vision-Language Large Models, ECCV'24
[3]FOLDER: Accelerating Multi-modal Large Language Models with Enhanced Performance,ICCV'25

---

> ### Author Rebuttal · Authors · 2025-07-31
>
> Thank you for your detailed review and thoughtful comments. Thank you for noting the strong motivation and the novelty of using visual–textual mutual information to adapt pruning to task complexity rather than relying on fixed, hand‑crafted policies.
>
> **Q1: Why the Budget-Constrained** **Logistic** **Retention is designed as Eq.5?**
>
> **A1:** We adopt a logistic retention function for three key reasons:
>
> 1. **Neuro‑inspiration:** As discussed in lines 208–210 of the main text, human visual reasoning over time follows an S‑shaped information accumulation curve, which closely matches the profile of a logistic function.
> 2. **Proven efficacy:** The logistic curve, often referred to as the logistic sigmoid, is extensively employed in contemporary LLM and VLM research. It underlies sigmoid-based language–image pre-training losses [1], serves as the gating function in mixture-of-experts routing [2], and plays a central role in various optimization algorithms [3].
> 3. **Empirical validation:** In Tab.4 (b), we compare linear, tanh, exponential, and logistic mappings. The logistic formulation consistently achieves the best balance between adhering to the token budget and maintaining task performance, justifying its selection.
>
> [1] Sigmoid Loss for Language-Image Pre-Training, ICCV 2023
>
> [2] Sigmoid Gating is More Sample Efficient than Softmax Gating in Mixture of Experts, NeurIPS 2024
>
> [3] Direct Preference Optimization: Your Language Model is Secretly a Reward Model, NeurIPS 2023
>
> **Q2: The reason for retention rate for easy and hard samples.**
>
> **A2:** We appreciate the reviewer’s insight. The observed retention patterns arise for three reasons:
>
> 1. **Fixed overall budget.** To satisfy our compute‑cost constraint, we enforce the same total token budget for both easy and hard samples. The curves therefore show the optimal pruning schedule under an identical compute budget.
> 2. **Behavior on easy samples.** Easy samples allow the model to rapidly locate in the relevant regions, so we can apply aggressive pruning in the earliest layers. After pruning, the model repeatedly attends to the retained tokens to guarantee correctness. (Note that if the compute budget were unconstrained, one could introduce further deep pruning to reduce computation even more for these easy cases.)
> 3. **Behavior on hard samples.** Difficult images require an initial exploratory phase in which the model must survey many tokens to locate the target. Hence, we delay pruning in early layers (high retention).
>
> **Q3: Improvements of experiments.**
>
> **A3:** We thank the reviewer for these valuable suggestions:
>
> 1. **More complex Image caption task for evaluation.**
>
>    We extend our pruning method to the COCO Caption dataset using the LLaVA 1.5-7B model and evaluate performance with the CIDEr and SPICE metrics. The unpruned baseline achieves a CIDEr score of 1.048. When retaining only 75%, 50% and 11.1% of the tokens, our approach continues to outperform existing methods on both metrics. Notably under heavy pruning at 11.1%, the gains are especially pronounced compared to others. These results underscore the strong generalization ability of our method.
>
>    | Method               | 75%       | 50%       | 11.1%     |
>    | -------------------- | --------- | --------- | --------- |
>    | FasterVLM（ICCV 25） | 1.032     | 0.997     | 0.872     |
>    | PyDrop (CVPR 25)     | 0.999     | 0.981     | 0.739     |
>    | **Ours**             | **1.039** | **1.028** | **0.946** |
>
> 2. **Inclusion of VTW, Turbo, and Folder.**
>
>    We appreciate the reviewer’s recommendation to examine VTW, Turbo and FOLDER. In Tab. 2 of the supplementary material, we include a comparison with VTW. We have conducted new experiments comparing AdaPrune with Turbo and FOLDER, and we will include a discussion of all three methods in our related‑work section. As shown in table below, AdaPrune achieves superior accuracy, again demonstrating its robustness and efficiency. All data in the table are taken directly from the original FOLDER publication.
>
>    | Method      | Retaining Ratio | MMBench-en↑ | MMMU↑    | MME↑     |
>    | ----------- | --------------- | ----------- | -------- | -------- |
>    | LLaVA1.5-7B | 100%            | 62.8        | 32.2     | 1339     |
>    | Turbo       | 34%             | 60.1        | 24.7     | 1302     |
>    | FOLDER      | 34%             | 61.4        | 31.3     | 1350     |
>    | **Ours**    | **34%**         | **64.3**    | **34.2** | **1433** |
>
> 3. **Absolute performance of compared baseline model.**
>
>    We have updated Tab. 3 to include the absolute scores of the baseline model (100% token retention). For example, the baseline accuracy is 41.49%, and AdaPrune achieves 46.15% at only 25% token usage. Including these numbers helps clarify that our relative gains correspond to a small absolute drop in accuracy while significantly reducing computation.
>
>    | **Method**      | **26/128 (20%)** | **32/128 (25%)** | **38/128 (30%)** | **45/128 (35%)** | **51/128 (40%)** |
>    | --------------- | ---------------- | ---------------- | ---------------- | ---------------- | ---------------- |
>    | **FasterVLM**   | 22.84            | 21.91            | 19.58            | 18.88            | 20.51            |
>    | **SparseVLM**   | 34.97            | 38.93            | 39.63            | 40.56            | 41.96            |
>    | **PyramidDrop** | 39.39            | 41.03            | 39.86            | 40.78            | 41.72            |
>    | **Ours**        | **40.09**        | **46.15**        | **44.29**        | **43.59**        | **43.36**        |

---

> > ### Comment · Reviewer_u9ow · 2025-08-07
> > **response to authors**
> >
> > Thanks for the response, I have raised my score.

---

> > > ### Author Response · Authors · 2025-08-07
> > > **Thank Reviewer u9ow for the raised score**
> > >
> > > We are grateful for your raised score. We will incorporate the additional results, elaborate on the technical details, and include a comparison with the new papers you mentioned.

---

> ### Author Response · Authors · 2025-08-04
>
> Thank you for your thoughtful comments and for the time you have invested in our work. We have posted our response and have successfully resolved the issues raised by the other two reviewers, who have in turn increased their scores. We hope that the above clarifications and the additional experiments sufficiently addressed your concerns. If any questions remain unresolved, please let us know. We would be grateful to clarify further and are happy to discuss specific points.

---

### Official Review · Reviewer_EN6i · 2025-07-03

**Clarity:** 3
**Significance:** 3
**Originality:** 2
**Rating:** 4
**Confidence:** 3

**Summary:**

The paper introduces AdaPrune, a training-free, plug-and-play framework for adaptive token pruning in Vision-Language Models. It measures cross-modal mutual information (MI) between early visual and textual tokens to gauge sample/task complexity, then maps MI to a budget-constrained logistic retention curve that dynamically allocates token budgets per decoder layer. Applied to LLaVA-1.5, LLaVA-NeXT, and a VLA model, AdaPrune prunes up to 89% of tokens while retaining >96% accuracy, outperforming prior methods.

**Questions:**

How exactly do the authors map the mutual information score to the logistic curve’s slope and midpoint? Did you apply any clamping or calibration on MI values?

It would be good to see AdaPrune on much larger VLMs or on streaming/video inputs. What challenges do the authors anticipate?

How sensitive is performance to noise in the MI estimate? Did you observe cases where mis-estimated MI led to under- or over-pruning?

It would be helpful to share more implementation details. For easier reimplementation, the authors can show more about how you compute MI from attention, your default hyperparameters, and how you pick which tokens to drop each layer?

**Ethical Concerns:**

["NO or VERY MINOR ethics concerns only"]

**Final Justification:**

I appreciate the authors' efforts spent on the rebuttal.
My main concerns are resolved, and I'm willing to keep a positive rating leaning to acceptance.

**Limitations:**

The paper omits an explicit “Limitations” section but combing it with the conclusion section. I think it is okay to do so.

**Quality:**

3

**Strengths And Weaknesses:**

Strengths:

The manuscript is well written and easy to follow.

AdaPrune uses a simple, training-free signal (mutual information between early visual and text tokens) to tailor pruning intensity for each input, avoiding manual schedule design or retraining.

The logistic retention curve formulation guarantees that any token or FLOPs budget is met exactly.

The authors conducted various experiments on LLaVA-1.5, LLaVA-NeXT, and a vision-language driving model.

---

Weaknesses:

The mapping from mutual-information scores to the curve’s slope and inflection point is described only at a high level, more detail on hyperparameter choices and token-selection criteria is needed.

Relying solely on an early-layer MI estimate risks misjudging complexity for inputs where initial attention patterns are misleading, potentially pruning critical tokens too soon.

Evaluation is limited to 7B-parameter models on static images. It is unclear how AdaPrune scales to much larger models, video inputs, or streaming scenarios. Table 4 in supp is helpful information, do you consider moving it to the main pages?

---

> ### Author Rebuttal · Authors · 2025-07-31
>
> Thank you for your detailed review and thoughtful comments. We appreciate your recognition of our clear writing, the simple training‑free MI signal that adapts pruning per input, the exact‑budget logistic retention curve, and the broad evaluations on LLaVA‑1.5/NeXT and a driving VLM.
>
> **Q1: More detail on** **hyperparameter** **choices and token-selection.**
>
> **A1:** Thank you for raising this important concern. Below we provide additional detail on our hyperparameter choices and token‑selection criteria:
>
> 1. **Hyperparameter** **mapping**
>
>    The mapping from mutual‑information score $I_q(V, T)$ to the slope $k_q$ and the inflection point $x_0^q$ is defined by
>
>    $$k_q = k_0 - \gamma\,I_q(V, T),\quad  x_0^q = x_0 + \beta\,I_q(V, T). $$
>
>    To demonstrate robustness, we fix these values across all experiments (See below for parameter selection). Specifically we set
>
>    $$k_0 = 0.4,\quad \gamma = 0.3,\quad x_0 = 15,\quad \beta = 0.3. $$
>
>    We apply only one clamp, enforcing $k_q \ge 0$ so that the selection process can only decrease the token number and never increase it. We do not apply any calibration to the mutual‑information values.
>
> 2. **Token‑selection criteria**
>
>    Visual tokens are scored by cross‑modal attention between image and text features, and we select the top $k$ tokens for downstream processing, with $k$ governed by our dynamic retention schedule. Our study targets dynamic control of the pruning budget rather than the design of ranking heuristics. To keep the analysis simple and comparable, we therefore use raw cross‑modal attention weights as the ranking criterion in all experiments. Although more sophisticated scoring mechanisms described in prior work may yield further gains, such enhancements fall outside the scope of the our study and are reserved for future investigation.
>
> 3. **Robustness across hyperparameters**
>
>    - **Parameter selection strategy. 1) Curve-based criteria.** As shown in Fig. 2, the retention curve must clearly distinguish easy from hard samples. We therefore choose parameters that provide sufficient dispersion, avoiding curves that concentrate around the midpoint or flatten at the extremes, so that the pruning schedule remains balanced. **2) Empirical** **validation** A small series of preliminary experiments yielded the following hyperparameters without extensive tuning:
>
>      $$k_0 = 0.4,\quad \gamma = 0.3,\quad x_0 = 15,\quad \beta = 0.3. $$
>
>      Specifying each value to one decimal place demonstrates that no overfitting occurred on any particular dataset.
>
>    - **More** **experiments.** To validate the robustness of these settings, we ran additional experiments under alternative hyperparameter configurations. Although some variants produced slightly improved results (see the table below), we elected to retain the original fixed values in order to ensure fairness and consistency across all datasets.
>
>      | Configuration                                      | TextVQA(token=64)↑ | TextVQA(token=128)↑ | GQA(token=64)↑ | GQA(token=128)↑ |
>      | -------------------------------------------------- | ------------------ | ------------------- | -------------- | --------------- |
>      | $k_0=0.3, \gamma=0.2, x_0=16, \beta=0.5$           | 57.2               | 57.6                | 57.3           | 59.4            |
>      | $k_0=0.5, \gamma=0.4, x_0=14, \beta=0.2 $          | 56.9               | 57.3                | 57.6           | 59.6            |
>      | $k_0=0.4, \gamma=0.3, x_0=15, \beta=0.3 $(default) | 57.1               | 57.4                | 57.7           | 59.9            |
>
> **Q2: Early-layer MI estimates risks misjudging.**
>
> **A2:** Thank you for raising this important and insightful concern.
>
> 1. **Optimal Layer Selection for MI Estimation.** Estimating mutual information too early can indeed introduce inaccuracies, which is why we perform MI estimation beginning at the second layer of the network. To address your question, we have conducted additional experiments measuring MI at various layers. The results, summarized in the table below, reveal two key trends. First, estimating MI too early (at layer 1) injects spurious signals into the pruning schedule, causing the model’s accuracy to suffer. Second, delaying MI estimation until very late layers front‑loads an excessive share of the fixed compute budget, which also diminishes overall performance. Estimating MI at layer 2 therefore achieves the optimal balance between reliable signal extraction and efficient budget utilization. Notably, in layers where MI is not estimated, we employ a linear pruning schedule in our experiments.
>
>    | **Layer** | TextVQA(token=64)↑ | TextVQA(token=128)↑ | GQA(token=64)↑ | GQA(token=128)↑ |
>    | --------- | ------------------ | ------------------- | -------------- | --------------- |
>    | **1**     | 56.4               | 56.8                | 56.5           | 59.1            |
>    | **2**     | 57.1               | 57.4                | 57.7           | 59.9            |
>    | **3**     | 56.9               | 57.2                | 57.4           | 59.5            |
>    | **4**     | 56.6               | 57.0                | 57.2           | 59.4            |
>
> 2. **Cross‑Modal Hybrid Token Scoring Mechanism.** Within the first layer, the fusion of textual and visual modalities remains insufficient. As shown in Sec. D of the supplementary materials, we use Cross‑Modal Weighted Pruning. We compute each visual token’s importance score by combining its self‑attention weight with its textual attention weight. This hybrid scoring prevents discarding tokens that carry critical visual content before stronger cross‑modal interactions form.
>
> **Q3: How AdaPrune scales to much larger models, video inputs, or streaming scenarios and Tab. 4 in supp.**
>
> **A3:** We appreciate the reviewer’s suggestion for scalability and future extensions to video and streaming.
>
> 1. **Scales to much larger models.**
>
>    AdaPrune is entirely model‑agnostic and requires no architectural modifications to operate on larger vision‑language models. In addition to the LLaVA‑13B results presented in Supplementary Table 4, we also evaluated AdaPrune on CogVLM-17B [1] by pruning 90 % of the tokens during inference. As shown in table below, AdaPrune consistently delivers significant inference‑cost reductions with negligible performance loss across both 7B, 13B and 17B variants, demonstrating robustness to model scale.
>
>    | Method               | Retaining Ratio | VQAv2↑   | VizWiz↑  | SQA↑     |
>    | -------------------- | --------------- | -------- | -------- | -------- |
>    | CogVLM-17B           | 100%            | 80.9     | 49.6     | 68.4     |
>    | FastV (ECCV 24)      | 10%             | 74.2     | 42.9     | 63.5     |
>    | FasterVLM（ICCV 25） | 10%             | 74.6     | 46.6     | 68.2     |
>    | **Ours**             | **10%**         | **75.7** | **48.3** | **68.4** |
>
> ​	[1] Cogvlm: Visual expert for pretrained language models, NeurIPS 2024
>
> 2. **Video and streaming inputs.** Thank you for this suggestion. Although our method is specifically designed for image–text VLMs, we believe its extension to video and real‑time streaming is highly promising and could bring significant benefits. The key challenge in this setting is maintaining temporal consistency in token selection, because pruning each frame independently may lead to flickering or unstable focus regions. We anticipate that simple extensions, such as (a) smoothing attention weights over successive frames to carry forward high‑importance tokens and (b) dynamically allocating a per‑frame token budget based on motion or scene change estimates, will address these issues with minimal changes to our framework. We plan to explore these temporal adaptations in future work.
>
> **Q4: Assessing AdaPrune’s Robustness to MI Estimation Noise.**
>
> **A4:** Thanks to the reviewer for raising the need to evaluate how MI estimation noise affects AdaPrune.
>
> We acknowledge that errors in the mutual information estimate can influence pruning decisions. When MI is overestimated, key tokens may be pruned too early. When MI is underestimated, the token set may be larger than necessary. Despite these effects, the impact on overall accuracy remains limited. Tab. 4(a) shows that a fixed logistic retention schedule ignoring per‑layer MI still achieves competitive performance.
>
> To measure sensitivity, we injected random perturbations of ±10 %, ±20 % and ±50 % into the MI values before computing the retention schedule and reran our experiments. The results show that perturbations in the 10–20% range cause only minor degradation in accuracy and only the large 50% noise produces a more noticeable drop. These findings confirm that AdaPrune remains robust, only degrading when MI estimation noise is substantial.
>
> |          | TextVQA(token=64)↑ | TextVQA(token=128)↑ | GQA(token=64)↑ | GQA(token=128)↑ |
> | -------- | ------------------ | ------------------- | -------------- | --------------- |
> | ±10 %    | 56.9               | 57.2                | 57.4           | 59.5            |
> | ±20 %    | 56.2               | 56.9                | 57.1           | 59.1            |
> | ±50 %    | 54.6               | 55.1                | 55.7           | 57.4            |
> | Original | 57.1               | 57.4                | 57.7           | 59.9            |
>
> **Q5: More implementation details of compute MI.**
>
> **A5:** We compute mutual information exactly as in Section 3.3 of the paper. Specifically, we first extract the attention map between the visual and textual modalities and normalize its entries to obtain the joint distribution $p(v_i, t_j)$. Treating the marginal distributions as $p(v_i)=\sum_j p(v_i,t_j)$ and $p(t_j)=\sum_i p(v_i,t_j)$, we then compute
>
> $$I(V, T) = \sum_{i=1}^{N_v}\sum_{j=1}^{N_t}  p(v_i, t_j)\,\log\frac{p(v_i, t_j)}{p(v_i)\,p(t_j)}. $$
>
> This directly measures the amount of information shared between visual and textual tokens based on their attention weights.

---

> > ### Comment · Reviewer_EN6i · 2025-08-06
> >
> > My main concerns are resolved, and I would like to thus increase the Significance score to 3.

---

> > > ### Author Response · Authors · 2025-08-06
> > > **Thank Reviewer EN6i for the increased significance score**
> > >
> > > We are grateful for your increased significance score. We will incorporate the additional results and technical details following your suggestions.

---

### Note · Authors · 2025-08-12

Dear ACs and Reviewers,

Thank you for your time reviewing our paper. We appreciate the positive advantages highlighted by all reviewers:

- Clear writing and a simple training-free framework that adapts pruning per input using cross-modal mutual information.
- Budget-constrained logistic retention that exactly meets token or FLOPs limits.
- Across LLaVA-1.5, LLaVA-NeXT, and a driving VLA, plus further results on larger models and tougher benchmarks added in the rebuttal, our method consistently delivers state-of-the-art performance.

During rebuttal, we address most reviewers' concerns.

- **EN6i**. Asked for mapping details, risks of early MI, sensitivity to MI noise, and implementation clarity. We provided the mapping, the clamp rule, the layer-2 choice with an ablation table, noise studies, and step-by-step MI computation from attention. **With the main concerns addressed, the reviewer raised the Significance score.**
- **u9ow**. Asked about the logistic choice, retention patterns across layers, broader tasks, related work, and absolute baselines. We justified the logistic mapping with neuro-inspired rationale and empirical wins, explained easy versus hard patterns under a fixed budget, added COCO Caption, added comparisons with VTW, Turbo, and FOLDER, and reported absolute baseline scores. **Reviewer raised the score.**
- **qWyB**. Requested explicit token-selection equations, MI layer choice across different depths, and hyperparameter search details. We confirmed exact attention aggregation used for ranking, committed to include it in the Method section, discussed fractional-depth generalization as future work, and specified the small, coarse search used to set hyperparameters. **With the doubts cleared, and the reviewer raised the score.**
- **zvmf**. Requested clearer writing, stronger motivation for simple versus complex patterns, applicability to recent models, more visualizations, and harder benchmarks. We will streamline prose, clarified the interpretation of Fig. 1, added results on Qwen2-VL and MMMU, and will include additional visualizations in the camera-ready. **With most concerns addressed, the reviewer updated the rating toward leaning accept.**

We thank the reviewers for the constructive guidance. We believe the concerns have been substantively addressed with new analyses and experiments, and the planned revisions will further improve clarity and completeness.

Best regards,

The authors of Paper 2958

---

### Decision · Program_Chairs · 2025-09-17

**Decision:**

Accept (poster)

**Comment:**

The authors propose an adaptive, training-free pruning strategy for large vision language models named AdaPrune, that tailors pruning policies to varying sample and task complexities. In short, AdaPrune quantifies the mutual information between visual and textual tokens, and then projects this signal to a budget-constrained logistic retention curve, quantifying the specific complexity of different tasks. This informs per-task pruning. Experiments on LLaVA-1.5, LLaVA-NeXT, and a vision-language driving model demonstrate the pruning performance of the proposed algorithm.


During the rebuttal phase, the authors provided a significant amount of new results, and details on their approach. They  evaluated AdaPrune on a larger VLM (CogVLM-17B) and to  Qwen2‑VL, extended their analysis to the MMMU benchmark, compared to Turbo and FOLDER, explored prunning performance over the COCO Caption dataset, and provided evaluations on the sensitivity to hyperparameters and the quality of the MI estimation. They also clarified how the  mutual information score is mapped the logistic curve, the scoring function used for token selection, how MI is estimated/the choice of the layer for MI estimation, and how hyperparameters were set. All of these should be added to the paper.

The same is true about the discussion on challenges when extending to streaming inputs, as well as the intuition provided w.r.t. certain design choices.